# Quantifying Myocardial Strain of the Left Ventricle in Normal People Using Feature-Tracking Based on Computed Tomography Imaging

**DOI:** 10.3390/diagnostics12020329

**Published:** 2022-01-27

**Authors:** Na Li, Tong Liu, Jia Liu, Yukun Cao, Yumin Li, Jie Yu, Xiaoyu Han, Guozhu Shao, Ming Yang, Zhihan Xu, Wenjuan Zeng, Heshui Shi

**Affiliations:** 1Department of Radiology, Union Hospital, Tongji Medical College, Huazhong University of Science and Technology, 1277 Jiefang Rd., Wuhan 430022, China; ncln1001@163.com (N.L.); Liutong_232022@163.com (T.L.); d201981614@hust.edu.cn (J.L.);yukuncao@hust.edu.cn (Y.C.); liyumin1215@163.com (Y.L.); gavinyuwhuh@hust.edu.cn (J.Y.); xiaoyuhan1123@163.com (X.H.); sgz15755376608@163.com (G.S.); ming.y@163.com (M.Y.); 2Hubei Province Key Laboratory of Molecular Imaging, Wuhan 430022, China; 3CT Collaboration, Siemens Healthineers Ltd., Shanghai 200126, China; zhihan.xu@siemens-healthineers.com; 4Department of Clinical Laboratory, Union Hospital, Tongji Medical College, Huazhong University of Science and Technology, Wuhan 430022, China

**Keywords:** computed tomography imaging, feature tracking, myocardial strain

## Abstract

Objective: The objective was to evaluate the normal value of left ventricular myocardial strain using the computed tomography feature-tracking technique and to explore the correlation between myocardial strains and cardiac function parameters. Methods: Participants suspected of coronary heart disease were selected from 17 August 2020 to 5 November 2020 to undergo coronary computed tomography angiography using a third-generation dual-source CT scanner. Data were imported into a commercial software (Medis) after multiphase reconstruction. The cardiac function parameters, radial (Err), circumferential (Ecc), and longitudinal strain (Ell) of the left ventricle were recorded. Results: A total of 87 normal subjects were enrolled, including 41 males and 46 females. For healthy subjects, the global radial strain (GRS), circumferential strain (GCS), and longitudinal strain (GLS) of the left ventricle were 74.5 ± 15.2%, −22.7 ± 3.0%, and −26.6 ± 3.2%, respectively. The Err and Ecc absolute values (|Ecc|) were the largest at the apex, and the |Ell| gradually increased from the base to the apex. The Err and |Ecc| were the largest in the lateral and inferior wall, respectively. |Ell| showed a clockwise decrease from the lateral wall in the short axis. Meanwhile, the GRS and |GLS| in females were higher than that in males. Multiple linear regression analysis showed that both SV and LVEF were the independent determinants of GRS, GCS, and GLS. BMI and CO were the independent determined factors of GCS. Conclusions: At a reasonable radiation dose, CT feature-tracking is a feasible and reproducible method to analyze left ventricular myocardial strain. Left ventricular myocardial strain in normal subjects varies in gender, segments, levels, and regions.

## 1. Introduction

Myocardial strain is defined as heart deformation during the systolic process, including radial strain (Err), circumferential strain (Ecc), and longitudinal strain (Ell) [1]. Feature-tracking (FT) is a highly reproducible way to measure myocardial strain, evaluating global and regional myocardial strain [2]. Myocardial strain of the left ventricle is an effective indicator to predict the outcomes of cardiovascular diseases, such as ischemic cardiomyopathy and non-ischemic cardiomyopathy [3]. Common methods to evaluate the systolic function include speckle-tracking imaging ultrasound and feature-tracking based on cardiac magnetic resonance imaging (FT-CMR). However, ultrasound (US) has a poor acoustic and depends on the operating angle and techniques. Furthermore, the time-consuming procedure of CMR in scanning limits its usage [4]. Computed tomography imaging (CT) is a rapid, convenient way to provides a one-stop evaluation of cardiac function when assessing coronary artery anatomy and is more standardized than US. Feature-tracking based on computed tomography (FT-CT) is an excellent way to assess cardiac function and is consistent with US and CMR [5,6,7]. However, many studies [3,8,9,10] have shown normal values of the left ventricular myocardial strain by US and CMR, whereas using FT-CT to evaluate the strain of the left ventricular myocardium in healthy subjects has rarely been reported.

Therefore, this study evaluates the normal values of the left ventricular myocardium strain by FT-CT. In addition, the correlations between myocardial strain and cardiac function parameters, gender, and age are analyzed to explore the feasibility and repeatability of the application of this technique, which could provide an important reference for clinical research and applications.

## 2. Materials and Methods

### 2.1. Study Population

An ethics committee institution of Tongji Medical College of Huazhong University of Science and Technology approved the study. People suspected of coronary heart disease (CAD) underwent coronary computed tomography angiography (CCTA) from 17 August 2020 to 5 November 2020, from which normal cases without coronary artery abnormality were selected. The exclusion criteria based on history were as follows: (1) electrocardiogram (ECG) abnormalities 2 weeks before the CT examination (T wave abnormalities, ST-T changes, and atrioventricular block); (2) coronary heart disease, cardiomyopathy, valvular heart disease, and other cardiovascular diseases; (3) history of revascularization; (4) history of hypertension, diabetes, or dyslipidemia. The exclusion criteria based on cardiac CT reports were as follows: (1) coronary artery calcification score >0; (2) coronary artery stenosis (≥1% lumen stenosis); (3) poor cardiac CT image quality.

### 2.2. Image Acquisition

CT was performed using a Siemens third-generation dual-source CT scanner (Somatom Force, Siemens Healthineers, Forchheim, Germany). Participants were trained to inhale and hold their breath before the examination. Retrospectively, the ECG-gated coronary computed tomography angiography (CCTA) scanning parameters were as follows: detector collimation 192 × 0.6 mm, gantry rotation time 0.25 s/r, pitch 0.15, and slice 0.75 mm. Automatic tube voltage technology (Care kV, Siemens Healthineers, Forchheim, Germany) and intelligent tube current scanning technology (Care Dose 4D, Siemens Healthineers, Forchheim, Germany) were used to automatically determine tube voltage and current. The reference tube voltage range was 100 kv, and reference tube current was 350 mAs. According to the weight of subjects, a total of 30–60 mL iopromide (400-mg I/mL; Bracco, Patheon Italia S. P. A, Ferentino, Italy) was injected into the median cubital vein, followed by diluted contrast and a saline flush. Diluted contrast was mixed saline solution with a ratio of 2:8, and the injection rate was 2–4 mL/s. The monitored region of interest was placed at the aortic root, and the trigger threshold of coronary artery CTA automatic scanning was set at 100 HU. When the density within the region of interest reached the threshold, scanning was automatically triggered, and the scanning time was 5–6 s. Twenty phases were reconstructed in 5% steps of the RR interval within the full window. The data constructive section thickness was 0.75 mm, the increment was 0.5 mm, the reconstruction kernel was Bv40, and the model-based iterative reconstruction (ADMIRE, Siemens Healthineers, Forchheim, Germany) was at a strength level of 3. The effective radiation dose was 4.0 ± 1.4 mSv (range: 1.9–6.7 mSv), equal to DLP (dose-length product) multiplied by 0.014.

### 2.3. Image Post-Processing and Analysis

The 20-phase images were imported to a commercial software package (Medis suite v3.0, Leiden, The Netherlands) to analyze the myocardial strain. Based on the American Heart Association 16-segment model for the left ventricle, the left ventricular basal, middle, and apex segments were selected for strain analysis. The end-diastole and -systole of the left ventricle was determined on the short- and long-axis sections, and the endocardial and epicardial borders were manually delineated, respectively. The deformation of myocardial movement was obtained by automatically tracking the continuous contour of the endocardial and epicardial borders throughout the cardiac cycle. In addition, the parameters of cardiac function were recorded, including the left ventricular end-diastolic volume (LVEDV), left ventricular end-systolic volume (LVESV), stroke volume (SV), cardiac output (CO), left ventricular ejection fraction (LVEF), and global and regional segment myocardial strain. The anterior wall of the left ventricle included segments 1, 2, 6, 7, 8, 12, and 13. The septal wall of the left ventricle included segments 2, 3, 8, 9, and 14. The inferior wall included segments 3, 4, 5, 9, 10, 11, and 15. The lateral wall included segments 5, 6, 11, 12, and 16. The Err and Ecc were measured in the short axis of the heart, and the Ell was an average value measured in the long axis (two, three, and four chambers) of the heart (Figure 1).

### 2.4. Repeatability

Thirteen subjects were randomly selected from the study’s total sample. The myocardial strain was measured independently by two radiologists with more than 5 years of experience in the diagnosis of cardiovascular diseases. In addition, one observer measured the myocardial strain after 2 weeks. The inter-observer and intra-observer interclass correlation coefficients (ICCs) of the global radial strain (GRS), global circumferential strain (GCS), and global longitudinal strain (GLS) were calculated separately to assess repeatability.

### 2.5. Statistical Analysis

Statistical analysis was performed using SPSS v21.0. The normality of the distributions for all continuous variables was tested using the Shapiro-Wilk test. The normal distribution data were described as the means ± standard deviation, whereas the non-normally distribution data were expressed as medians (interquartile ranges) and categorical variables were expressed as frequencies (percentages), respectively. Independent sample student’s *t*-tests were used to compare two groups of normally distributed variables. One-way analysis of variance was used to compare multiple groups. Furthermore, Bonferroni and Tamhanes were used for post-hoc comparisons between the two groups. Pearson’s and Spearman correlation coefficients were implemented for evaluating correlations between continuous variables as appropriate. In addition, multivariable linear regression was further used to analyze the relationship between the parameters and myocardial strain. Finally, repeatability between observers was evaluated with 13 randomly selected patients using ICCs. *p* < 0.05 (two-tailed) was considered statistically significant.

## 3. Results

### 3.1. Clinical Indicators and Cardiac Function Parameters of the Study Population

In this study, 827 patients with basic clinical information and CCTA examination data were included, and 87 healthy subjects were eventually enrolled (Figure 2), including 41 males and 46 females, aged 27–79 years (48 ± 11 years old). The patients were divided into three groups according to age: young (≤40 years old), middle-aged (40–55 years old), and elderly (>55 years old). There were statistically significant differences in the age, height, weight, and body mass index (BMI) between males and females, respectively (*p* < 0.05). Cardiac function parameters, including LVEDV, LVESV, SV, LVEF, CO, and heart rate (HR), did not differ between the sexes, respectively (*p* > 0.05) (Table 1).

### 3.2. Global and Regional Myocardial Strain of the Left Ventricle

In this study, 4176 strain values of myocardial segments were measured. After excluding segments with poor image quality and abnormal local myocardial strain due to severe myocardial bridge, 4134 values were statistically analyzed. The global and segment myocardial strain values of the left ventricle are shown in Figure 3.

The myocardial strain in different parts of the left ventricle is shown in Table 2. The Err of the middle segment of the left ventricle was lower than that of the basal and apex segments. |Ecc| (the absolute value of Ecc) in the apical segment was higher than that in the basal and middle segments, and the base segment was higher than that in the middle segment. |Ell| (the absolute value of Ell) gradually increased from the basal to the apex segment.

The myocardial strain in different ventricular walls of the left ventricle is shown in Table 3. In the lateral wall of the left ventricle, Err and |Ell| were the largest, and |Ecc| was the smallest. Err showed as the lateral wall > inferior wall > anterior wall > septal wall. |Ecc| showed as the inferior wall > septal wall > anterior wall > lateral wall. |Ell| showed a clockwise decreasing trend from the lateral wall in the short axis (Table 3).

### 3.3. The Relationship between Left Ventricular Myocardial Strain and Clinical Characteristics

There were differences in the left ventricular GRS, GLS, and GCS based on the gender of the participant (Table 4). The GRS, |GCS|, and |GLS| (the absolute values of GCS and GLS) in females were higher than those in males (GRS: (77.7 ± 14.7)% vs. (70.9 ± 15.1)%, *p* < 0.05; GLS: (−27.3 ± 2.6)% vs. (−25.8 ± 3.6)%, *p* < 0.05; GCS: (−23.0 ± 3.1)% vs. (−22.3 ± 2.9)%, *p* = 0.286). The differences in myocardial strain on segment and region of the left ventricle in gender are shown in Figure 4.

The correlation between left ventricular myocardial strains, age, and BMI is shown in Figure 5. GRS was significantly positively correlated with age (r = 0.219, *p* < 0.05). While GCS and GLS were negatively correlated with age respectively (r = −0.127, *p* = 0.242; r = −0.209, *p* = 0.052). GCS and GLS were significantly positively correlated with BMI respectively (r = 0.274, *p* < 0.05; r = 0.223, *p* < 0.05). In addition, LVEDV and LVESV were negatively correlated with age (r = −0.222, *p* < 0.05; r = −0.274, *p* < 0.05), respectively. There was no significant difference in the GRS, GCS, and GLS among all age groups (Figure 6).

### 3.4. The Relationship between Left Ventricular Myocardial Strain and Left Ventricular Function Parameters

The correlation between left ventricular myocardial strains and left ventricular function parameters is shown in Figure 7. The GRS of the left ventricle was significantly negatively correlated with LVESV (r = −0.256, *p* < 0.05) and significantly positively correlated with LVEF (r = 0.632, *p* < 0.01). GCS was significantly positively correlated with LVESV (r = 0.459, *p* < 0.01) and significantly negatively correlated with SV, LVEF, and CO (r = −0.432, −0.831, and −0.323, respectively, *p* < 0.01). There was a significantly negative correlation with GLS, SV, and LVEF (r = −0.326 and −0.416, respectively, *p* < 0.01 in each).

Furthermore, univariate and multivariate regression analysis for global myocardial strains and clinical indicators and cardiac function parameters are shown in Table 5. Both the SV (GRS: β = 0.481, *p* = 0.004; GCS: β = −0.115, *p* < 0.001; GLS: β = −0.14, *p* = 0.003) and LVEF (GRS: β = 1.179, *p* < 0.001; GCS: β = −0.345, *p* < 0.001; GLS: β = −0.13, *p* = 0.01) were the independent determinants of GRS, GCS, and GLS. The GRS of females was 8.174% higher than males, and the GLS was −1.653% lower than males. BMI (β = 0.255, *p* < 0.001) and CO (β = 0.854, *p* = 0.022) were the independent determined factors of GCS.

### 3.5. Reproducibility

The ICC values in the intra-observer analysis were 0.575, 0.900, and 0.891 for GRS, GCS, and GLS, respectively. Finally, the ICCs were 0.325, 0.853, and 0.787 for GRS, GCS, and GLS, respectively, in the inter-observer analysis (Table 6).

## 4. Discussion

It is important to understand the normal global structural and functional parameters of the left ventricle, which is the basis for evaluating cardiovascular diseases. Furthermore, it is worth applying a measurement technique in the research of reference values, as this helps the timely clinical discovery of abnormalities, diagnosis, and treatment.

The normal values of the left ventricular myocardial strains quantified by FT-CT reveal significant differences in gender and segments and correlate with clinical characteristics and left ventricular function. This study shows that the GRS, GCS, and CLS of the left ventricle were 74.5 ± 15.2%, −22.7 ± 3.01%, and −26.6 ± 3.2%, respectively, in healthy individuals. The absolute values measured by FT-CT were higher than the normal values measured by Cao [11] and Liu [4] using the Cvi and Trufi Strain software, respectively. The differences might be related to the scanning equipment and implemented measurement methods. It has been proven that there are differences in measurement methods among different software [2]. The myocardial FT technique is based on identifying features in images and tracking them in continuous images [12], depending on the quality of imaging and sharpness of the endocardial boundary. CT examination has high spatial and contrast resolution, and less influence on the myocardium border than CMR, which is caused by blood flow artifacts on the left ventricle. These may be the reasons why the values measured by FT-CT were higher than those measured by FT-CMR.

Our study shows that there were differences in the three short axes of the left ventricular regional myocardial strains. The Err and Ecc of the left ventricle were the highest in the apex and lowest in the middle, which agrees with the results of Ecc measured by Taylor [10] using FT-CMR. However, a different result has also been reported, which showed that Ecc was highest in the apex and lowest in the basal measured by ultrasonic speckle-tracking imaging [13]. This may result from the imaging technology. In contrast, the absolute value of Ell, showing a gradually increasing trend from the base, agrees with the results reported by Leitman [13] and Qian [9].

In addition, significant differences in regional myocardial strains were observed among different segments and walls of the left ventricle, which agrees with previous studies [14,15,16]. This is the expression of local non-uniformity of the left ventricular function in healthy people, showing differences in myocardial strain among different short axis levels and ventricular walls. The literature suggests that the left ventricle has a high regional morphological and functional non-uniformity in healthy individuals, including transmural, apex to base, and circumferential non-uniformity. This shows heterogeneity in the synchronous contraction of the heart and may impact the effectiveness of regional functional assessments [17,18]. The segmental radial strains vary greatly, and the segmental circumferential and longitudinal strains show a relatively consistent normal range. Therefore, it is necessary to quantify the segmental specific reference value for the circumferential and longitudinal strains in healthy individuals.

It has been reported that left ventricular myocardial strains partly differ between genders and age groups in healthy subjects [3,10,19]. The study found that the GRS and GLS values of healthy females were higher than those of males, which disagrees with the study by Andre et al. [3], which found that the GRS in women is lower than that in men. In their study, there was a significant difference in the circumferential strain between genders, whereas only one trend was shown in our sample, which may be related to the relatively small sample size, population age, and BMI in our study. However, the differences in the GRS and GCS between genders in our study agree with those of other studies [10,15]. First, there were differences in the volume, mass, and the LVEF of the left ventricle, which is normal between males and females, and differences in myocardial deformation [20,21]. In this study, the GRS was positively correlated with age, and the age of female subjects was greater than that of the male subjects. Furthermore, the GLS was positively correlated with BMI, and the BMI of male samples was greater than that of female subjects. Therefore, the GRS and GLS values of healthy females in the study were higher than those of males. The analysis of the differences in myocardial systolic peak strain between genders can help further research of cardiac movement.

In addition, this study found a significant positive correlation between GRS and age, and a negative correlation between LVEDV, LVESV, and age. The literature shows that with increasing age, LVEDV shrinks significantly and myocardial mass decreases, leading to a significant increase in the left ventricular (LV) mass/LVEDV ratio. Then, LV remodeling occurs, with an increase in myocardial stiffness and a decrease in compliance and diastolic function, which results in a compensatory thickening of the LV myocardium during systole, eventually leading to the increase of GRS [22]. Meanwhile, the study showed a significant positive correlation between GRS and LVEF, and a significant negative correlation between GCS, GLS, and LVEF. An increase in myocardial strain indicates stretching of the LV myocardial fibers, myocardial thickening, and increased myocardial contractility, with a corresponding increase in LVEF. Conversely, a decrease in myocardial strain indicates the shortening of myocardial fibers, myocardial thinning, and a corresponding decrease in LVEF [18]. Thus, it may provide incremental diagnostic value when studying the ominous decrease in ejection fraction after systolic heart failure or myocardial infarction using a comprehensive assessment of LV systolic function with myocardial strain.

Another significant positive correlation was revealed between left ventricular GCS, GLS, and BMI in healthy individuals, and similar results have not been reported in the literature. Whether GCS and GLS based on cardiac CT can become early warning indicators to quantify the health status of obese people remains to be studied.

In this study, GCS and GLS had good intra-observer and inter-observer reproducibility, and GRS measurements were less stable. The poor reproducibility of the radial parameters may be caused by the quantification of the radial strain, which depends on the simultaneous motion of the endocardium and epicardium. Furthermore, the density contrast at the epicardial border is less pronounced than at the endocardial border. Moreover, there are more myocardial trabeculae in the apical region. The compression and drainage of blood from the trabeculae at the end of systole alter the voxel appearance in this region, making accurate tracking challenging [19]. Finally, when the blood space between the trabeculae closes during systole, the border between the trabeculae and the dense portion of the myocardium may move, resulting in blurring of the endocardial contours.

This study has some limitations. First, it is a single-center, single-race study. Therefore, the sample size of males and females were unequal, and there is a possibility that some gender differences were magnified or inadequately detected. Considering the limited overall sample size and the number of samples from youth and elderly groups, the results of myocardial strain between groups stratified by age may require further analysis by expanding the sample size. Second, this study did not perform a concordance test between myocardial strain measured using CT FT techniques and other imaging techniques. However, previous studies have shown good concordance between CT FT techniques, ultrasound speckle-tracking techniques, and FT-CMR to assess myocardial strain in patients with cardiovascular disease. Lastly, the radiation dose was relatively high due to the use of a retrospective electrocardiogram gated scan. However, we used techniques (CarekV and CareDose) to reduce the radiation dose and ensure a one-stop evaluation of coronary anatomy and myocardial strains at a comparable dose. In addition, compared with the radiation dose described by the authors of [23] and the diagnostic reference level of an international cardiac CT [24] (DLP was 348 mGy·cm, 400 mGy·cm, respectively), our average effective radiation dose was reasonable and was not higher than theirs (average effective radiation dose: 4.0 ± 1.4 mSv).

## 5. Conclusions

Conclusively, using the CT FT technique to assess left ventricular myocardial strain in healthy individuals is feasible and reproducible. In the future, the overall sample size can be increased to quantify the reference values of LV myocardial strain in different age groups. Furthermore, given the rapidity and convenience of CT examination, using CT to quantitatively assess myocardial strain differences by gender and segment should improve the quantitative diagnosis of cardiovascular diseases, such as coronary heart disease and cardiomyopathy.

## Figures and Tables

**Figure 1 diagnostics-12-00329-f001:**
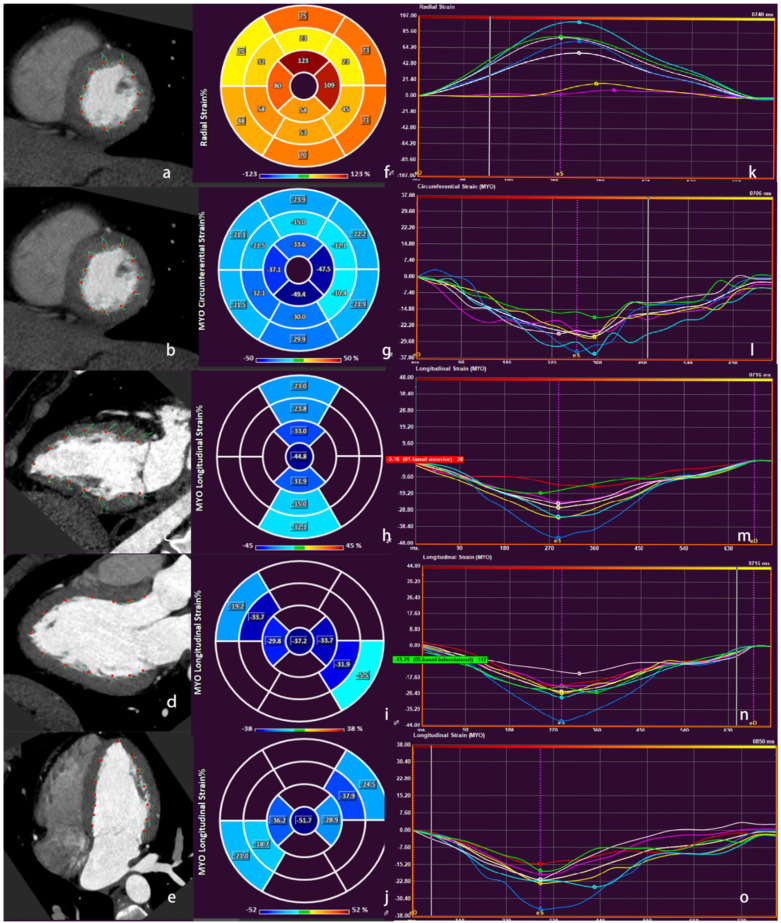
Schematic diagram of myocardial strain on post-processing software. (**a**–**e**) Images of different planes of the cardiac; (**f**–**j**) images of Err, Ecc and Ell on the short axis of the cardiac; (**k**–**o**) curves of myocardial strain and time in the cardiac cycle. Err: radial strain; Ecc: circumferential strain; Ell: longitudinal strain.

**Figure 2 diagnostics-12-00329-f002:**
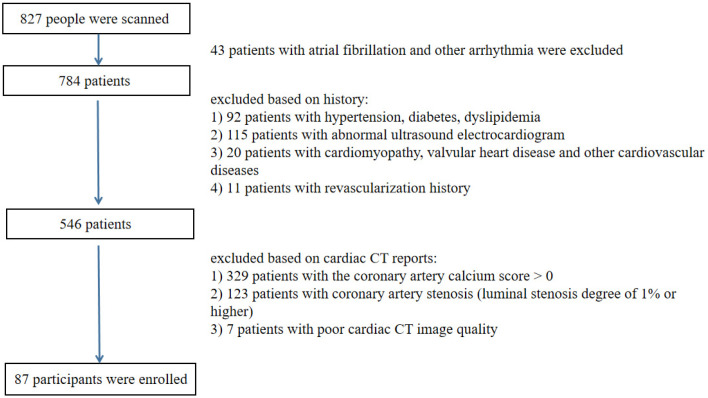
Flowchart of patients who met the inclusion/exclusion criteria for the study.

**Figure 3 diagnostics-12-00329-f003:**
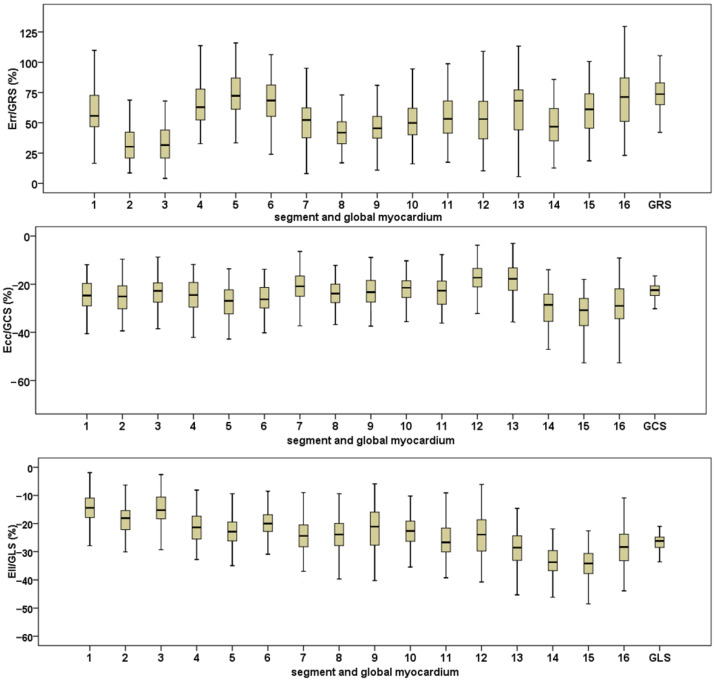
The boxplots for the global and segment myocardial strain value. The top is the image of about 16 segments and the global radial strain. The middle is the image of about 16 segments and the global circumferential strain. The bottom is the image of about 16 segments and the global longitudinal strain. The center box of the boxplots indicates the values from the lower quartile to the higher quartile. The center line shows the median. The range of whiskers is from the minimum to the maximum value. GRS: global radial strain; GCS: global circumferential strain; GLS: global longitudinal strain. Err: radial strain; Ecc: circumferential strain; Ell: longitudinal strain.

**Figure 4 diagnostics-12-00329-f004:**
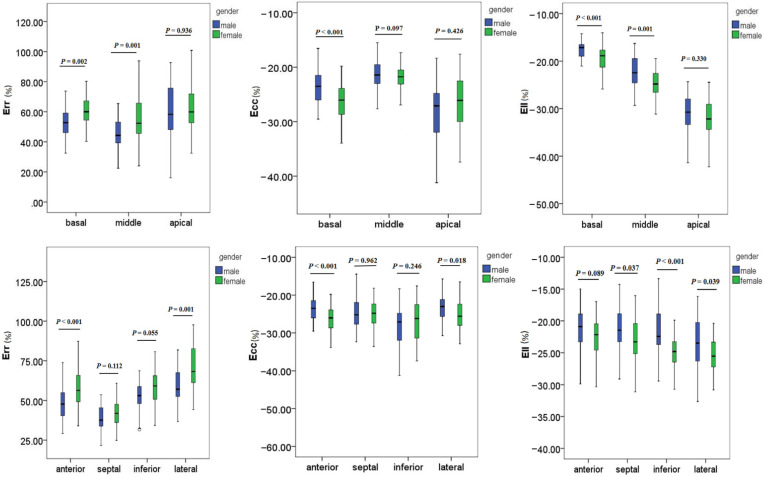
The boxplots for myocardial strain on the level and region of the left ventricular in different genders. The center box of the boxplots indicates the values from the lower quartile to the higher quartile. The center line shows the median. The range of whiskers is from the minimum to the maximum value. Err: radial strain; Ecc: circumferential strain; Ell: longitudinal strain.

**Figure 5 diagnostics-12-00329-f005:**
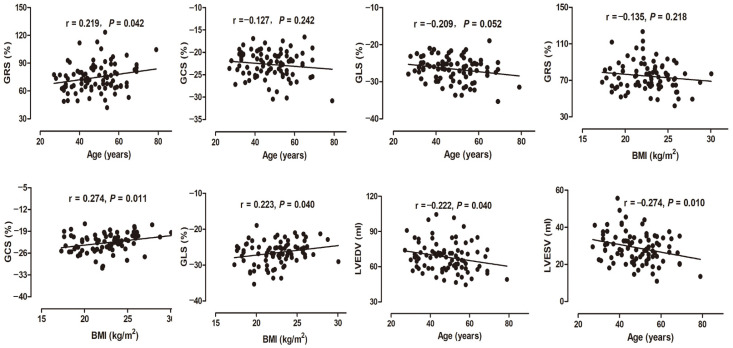
The correlation between left ventricular strain with age and BMI, and the correlation between LVEDV and LVESV with age. GRS: global radial strain; GCS: global circumferential strain; GLS: global longitudinal strain. LVEDV: left ventricular end-diastolic volume; LVESV: left ventricular end-systolic volume; BMI: body mass index.

**Figure 6 diagnostics-12-00329-f006:**
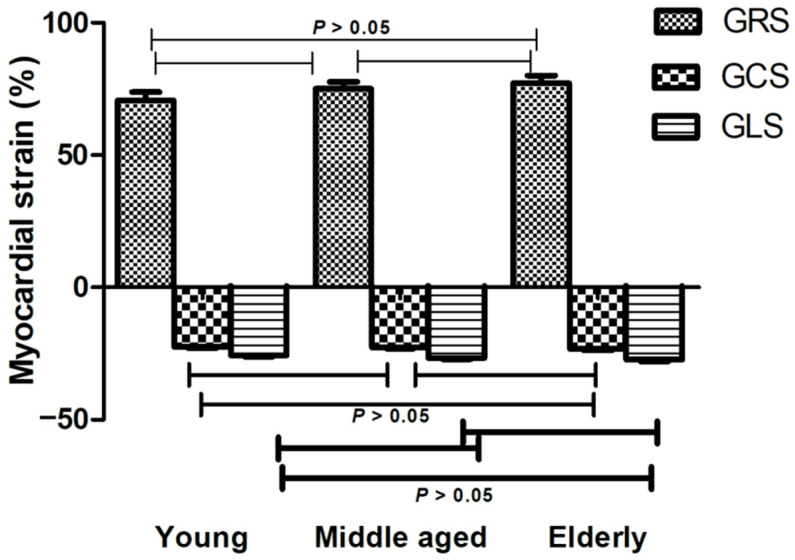
Myocardial strain in different age groups. There was no significant difference in global myocardial strain among all age groups. GRS: global radial strain; GCS: global circumferential strain; GLS: global longitudinal strain.

**Figure 7 diagnostics-12-00329-f007:**
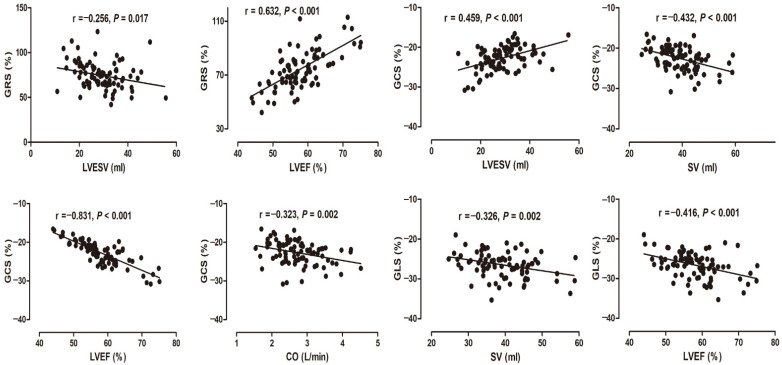
The correlation analysis between left ventricular global myocardial strain and left ventricular function parameters. GRS: global radial strain; GCS: global circumferential strain; GLS: global longitudinal strain. LVEDV: left ventricular end-diastolic volume; LVESV: left ventricular end-systolic volume; SV: stroke volume; CO: cardiac output; LVEF: left ventricular ejection fraction.

**Table 1 diagnostics-12-00329-t001:** Clinical indicators and cardiac function parameters of the study population.

Variable	Males (*n* = 41)	Females (*n* = 46)	Total (*n* = 87)	*p* Values
Age group				
≤40 years old	16	7	23	-
40–55 years old	17	23	40	-
>55 years old	8	16	24	-
Age (years)	44.3 ± 10.9	51.4 ± 10.3	48.1 ± 11.1	0.030 *
Height (cm)	169.9 ± 5.3	160.5 ± 4.9	165.0 ± 6.9	<0.001 *
Weight (kg)	67.3 ± 9.2	56.2 ± 7.1	61.6 ± 9.9	0.001 *
BMI (kg/m^2^)	23.3 ± 2.6	21.8 ± 2.7	22.5 ± 2.7	0.014 *
HR((beats/min)	66.7 ± 11.3	70.6 ± 13.1	68.7 ± 12.3	0.155
Cardiac function parameters				
LVEDV (mL)	71.0 ± 14.6	65.7 ± 9.9	68.2 ± 12.5	0.050
LVESV (mL)	30.5 ± 9.6	27.7 ± 6.7	29.0 ± 8.3	0.125
SV (mL)	40.8 ± 8.7	37.9 ± 5.8	39.2 ± 7.4	0.084
LVEF (%)	57.7 ± 6.9	58.1 ± 6.5	57.9 ± 6.6	0.752
CO(L/min)	2.8 ± 0.7	2.7 ± 0.6	2.7 ± 0.6	0.643

Note: All data are expressed as the mean ± SD, and number of participants (without percentages); BMI: body mass index; HR: heartbeat; LVEDV: left ventricular end-diastolic volume; LVESV: left ventricular end-systolic volume; SV: stroke volume; CO: cardiac output; LVEF: left ventricular ejection fraction. * *p* < 0.05 between groups.

**Table 2 diagnostics-12-00329-t002:** Myocardial strain in different parts of the left ventricle.

	The Basal	The Middle	The Apical	*p* Values
Err (%)	57.0 ± 12.2	51.2 ± 15.7	62.7 ± 22.1 ^b^	<0.001
Ecc (%)	−24.9 ± 3.6	−21.8 ± 3.0 ^a^	−27.4 ± 5.9 ^a,b^	<0.001
Ell (%)	−18.6 ± 2.8	−23.8 ± 3.5 ^a^	−31.6 ± 4.0 ^a,b^	<0.001

Note: All data are expressed as the mean ± SD. Err: radial strain; Ecc: circumferential strain; Ell: longitudinal strain. ^a^: *p* < 0.0167 compared with the basal segment; ^b^: *p* < 0.0167 compared with the middle segment.

**Table 3 diagnostics-12-00329-t003:** Myocardial strain in different ventricular walls of the left ventricle.

	Anterior	Septal	Inferior	Lateral	*p* Values
Err (%)	54.0 ± 12.2	41.2 ± 9.2 ^a^	56.3 ± 10.6 ^b^	66.1 ± 15.8 ^a,b,c^	<0.001
Ecc (%)	−24.9 ± 3.6	−24.9 ± 3.8	−27.4 ± 5.9 ^a,b^	−24.4 ± 4.0 ^c^	<0.001
Ell (%)	−22.1 ± 3.5	−22.4 ± 4.0	−23.3 ± 3.6	−24.5 ± 4.3 ^a,b^	<0.001

Note: All data are expressed as the mean ± SD. Err: radial strain; Ecc: circumferential strain; Ell: longitudinal strain. ^a^: *p* < 0.008 compared with anterior wall; ^b^: *p* < 0.008 compared with septal wall; ^c^: *p* < 0.008 compared with inferior wall.

**Table 4 diagnostics-12-00329-t004:** Global myocardial strain of the left ventricle in different genders.

	Males	Females	*p* Values
GRS (%)	70.9 ± 15.1	77.7 ± 14.7	0.038 *
GCS (%)	−22.3 ± 2.9	−23.0 ± 3.1	0.286
GLS (%)	−25.8 ± 3.6	−27.3 ± 2.6	0.023 *

Note: All data are expressed as the mean ± SD. GRS: global radial strain; GCS: global circumferential strain; GLS: global longitudinal strain. * *p* < 0.05.

**Table 5 diagnostics-12-00329-t005:** Univariate and multivariate regression analysis for global myocardial strains and clinical indicators and cardiac function parameters.

Variable	GRS		GCS		GLS	
	Univariate Analysis		Univariate Analysis		Univariate Analysis	
	*β* Value	*p* Value	*β* Value	*p* Value	*β* Value	*p* Value
Age (years)	0.299	**0.042**	−0.034	0.242	−0.060	0.052
Female *	6.756	**0.038**	−0.694	0.286	−1.550	**0.023**
Height (cm)	−0.119	0.626	−0.016	0.737	−0.010	0.842
Weight (kg)	−0.208	0.427	0.057	0.088	0.050	0.160
BMI (kg/m^2^)	−0.762	0.218	0.304	**0.011**	0.264	**0.040**
HR (beats/min)	−0.071	0.611	0.096	0.389	0.048	0.098
LVEDV (mL)	−0.014	0.913	0.019	0.477	−0.017	0.548
LVESV (mL)	-0.472	**0.016**	0.167	**<0.001**	0.063	0.131
SV (mL)	0.738	**<0.001**	−0.170	**<0.001**	−0.138	**0.002**
LVEF (%)	1.448	**<0.001**	−0.378	**<0.001**	−0.201	**<0.001**
CO (L/min)	9.282	**<0.001**	−1.567	**0.002**	−0.770	0.167
**variable**	**Multivariate analysis**		**Multivariate analysis**		**Multivariate analysis**	
	***β*** **value**	***p*** **value**	***β*** **value**	***p*** **value**	***β*** **value**	***p*** **value**
Female *	8.174	**<0.001**	-	-	−1.653	**0.010**
BMI (kg/m^2^)	-	-	0.255	**<0.001**	-	-
SV (mL)	0.481	**0.004**	−0.115	**<0.001**	−0.14	**0.003**
LVEF (%)	1.179	**<0.001**	−0.345	**<0.001**	−0.13	**0.010**
CO (L/min)	-	-	0.854	**0.022**	-	-

Note: * Males as reference; GRS: global radial strain; GCS: global circumferential strain; GLS: global longitudinal strain. BMI: body mass index; HR: heartbeat; LVEDV: left ventricular end-diastolic volume; LVESV: left ventricular end-systolic volume; SV: stroke volume; CO: cardiac output; LVEF: left ventricular ejection fraction. Stepwise linear regression was used to analyze multivariate regression.

**Table 6 diagnostics-12-00329-t006:** The results of reproducibility in global myocardial strains.

	Intra-Observer		Inter-Observer	
	ICCs	CV	ICCs	CV
GRS	0.575	5.0%	0.325	1.9%
GCS	0.900	0.2%	0.853	2.1%
GLS	0.891	1.2%	0.787	1.4%

Notes: GRS: global radial strain; GCS: global circumferential strain; GLS: global longitudinal strain; ICC: intraclass correlation coefficients; CV: coefficient of variation.

## Data Availability

The data presented in this study are available on request from the corresponding author.

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
