# Peer review of "Quantifying Myocardial Strain of the Left Ventricle in Normal People Using Feature-Tracking Based on Computed Tomography Imaging"

_diagnostics, 2022, doi:10.3390/diagnostics12020329_

Round 1

Reviewer 1 Report

Nice study and well written. However, you should include the evident shortcomings of this method (point 8).

1.abstract: Maybe "healthy" is not the correct term for patients undergoing CCTA, as obviously there was a clinical indication for the CT
2. abstract, results: I recommend removing brackets for the results
3. As you discuss shortcomings of US and CMR for deature-tracking you should also discuss drawbacks of CT for feature-tracking (radiation, temporal resolution) in the introducation or discussion
4. Methods/Results. Please give the indicatins for CCTA for your patient cohort
5. Methods: Which reference values did you choose for Carekv and CareDose?
6. Methods: There was automatic tracking of endocardial and epicardial borders if I understand correctly. Did you have to manually correct in some cases? Please give numbers and state whether endocardial or epicardial contours were corrected and why.
7. Results: How long id it take to evaluate strains? 
8. Discussion: The problem of CT-feature tracking is that retrospective ECG-gating is necessary without dose modulation. That means tht the radiation dose is unnecessarily high. You should definitely discuss this limitation of CT-FT. This also limits the statement of your conclusion. Therefore, please modify your conclusion

Author Response

Response to comments

Point 1: ---  abstract: Maybe "healthy" is not the correct term for patients undergoing CCTA, as obviously there was a clinical indication for the CT . 

Response: We sincerely thank your insightful comments, and we have changed “healthy” to “normal” in the revised manuscript.

point 2: ---  abstract, results: I recommend removing brackets for the results.
Response: We deeply thank your kind comments, and we have removed the brackets for the result in abstract of the revised manuscript.

point 3: ---  As you discuss shortcomings of US and CMR for feature-tracking you should also discuss drawbacks of CT for feature-tracking (radiation, temporal resolution) in the introduction or discussion.

Response: We highly appreciate your insightful comment. We have discussed the drawbacks of CT for feature-tracking in discussion.

point 4: ---  Methods/Results. Please give the indications for CCTA for your patient cohort.

Response: We deeply thank your kind comments. This is a retrospective study and the patients performed CCTA was suspected of CAD (symptoms such as chest tightness, chest pain, etc. ), and we have added the indications ( suspected of CAD ) into the study population.

point 5: ---  Methods: Which reference values did you choose for Carekv and CareDose?

Response: We deeply thank your kind comments. We have added the reference values to the image acquisition of revised manuscript: The reference tube voltage was 100 KV and reference tube current was 350 mAs.

point 6: ---  Methods: There was automatic tracking of endocardial and epicardial borders if I understand correctly. Did you have to manually correct in some cases? Please give numbers and state whether endocardial or epicardial contours were corrected and why.
Response: We deeply thank your questions. The software can automatically recognize and track the endocardial and epicardial border. But when the image has slight respiratory motion artifacts, which may interfere with the software's automatic recognition of the borders tracking, manual correction is needed. So, the patient would be trained to inhale and hold their breath before the examination. Less than 10 cases underwent manual correction. 

point 7: ---  Results: How long id it take to evaluate strains? 
Response: We deeply thank your questions. About 5 to 10 minutes will be taken to measure one case. The time is short when the software automatically identifies and outlines accurately, and it’s relatively long when manual correction is required.

Point 8: ---  Discussion: The problem of CT-feature tracking is that retrospective ECG-gating is necessary without dose modulation. That means that the radiation dose is unnecessarily high. You should definitely discuss this limitation of CT-FT. This also limits the statement of your conclusion. Therefore, please modify your conclusion.
Response: We highly appreciate your insightful comment. The radiation dose was relatively high because of using a retrospective electrocardiogram gated scan. But we use techniques (CarekV and CareDose) to reduce the radiation dose and ensure a one-stop evaluation of coronary anatomy and myocardial strains at comparable dose. And compared with the radiation dose in the paper[1] and diagnostic reference level of an international cardiac CT[2] (DLP was 348 mGy·cm and 400 mGy·cm respectively), our average effective radiation dose are reasonable and was not higher than theirs (average effective radiation dose: 4.0±1.4 mSv). Thank you again for the suggestions and we discussed this part in Limitations. The conclusions have been modified.

Reference:

  1. Yoshida K, Tanabe Y, Kido T,et al. Characteristics of the left ventricular three-dimensional maximum principal strain using cardiac computed tomography: reference values from subjects with normal cardiac function[J]. Eur Radiol. 2020, 30:6109-6117.
  2. Stocker TJ, Deseive S, Leipsic J, et al. Reduction in radiation exposure in cardiovascular computed tomography imaging: results from the PROspective multicenter registry on radiaTion dose Estimates of cardiac CT angIOgraphy iN daily practice in 2017 (PROTECTION VI)[J]. European Heart Journal, 2018,39(41):3715-3723.

Modified: 

---  In limitations:

We have added “ lastly, The radiation dose was relatively high because of using a retrospective electrocardiogram gated scan. But we use techniques (CarekV and CareDose) to reduce the radiation dose and ensure a one-stop evaluation of coronary anatomy and myocardial strains at comparable dose. And compared with the radiation dose in the paper[23] and diagnostic reference level of an international cardiac CT[24](DLP was 348 mGy·cm and 400 mGy·cm respectively), our average effective radiation dose are reasonable and was not higher than theirs(average effective radiation dose: 4.0±1.4 mSv).”

---  In conclusions of abstract:

The conclusions were modified as “At reasonable radiation dose, CT feature-tracking is a feasible and reproducible method to analyze left ventricular myocardial strain. Left ventricular myocardial strain in healthy subjects varies in gender, segments, levels, and regions.”

Reviewer 2 Report

The study is of interest nevertheless, I have two comments: 

  • Despite of limitations, if you have additional data at least of ultrasounds (or CMR), please integrate the results with comparison test between myocardial strain measured using CT FT and other imaging techniques.
  • Please insert a table to show inter-observer variability through interclass correlation coefficients or K Cohen.

Author Response

Response to comments

point 1:

---  Despite of limitations, if you have additional data at least of ultrasounds (or CMR), please integrate the results with comparison test between myocardial strain measured using CT FT and other imaging techniques.
Response: We deeply thank your constructive suggestion. While it is sorry that we did not have any data of ultrasounds or CMR for myocardial strain calculation due to the retrospective study. This was also mentioned in our manuscript at Limitation section. While, as we mentioned in the manuscript, numerous studies have proven the good agreements between the CT derived myocardial strain and ultrasound or CMR strain. Hence, our primary purpose of this study is to provide a reference value for CT myocardial strain in normal subjects. 

point 2:

---  Please insert a table to show inter-observer variability through interclass correlation coefficients or K Cohen.
Response:We deeply thank your kind suggestions. We have added a table in the reproducibility of result. The table shows as follows:

Table 6. The results of reproducibility in global myocardial strains.

intra-observer

inter-observer

ICCs

CV

ICCs

CV

GRS

0.575

5.0%

0.325

1.9%

GCS

0.900

0.2%

0.853

2.1%

GLS

0.891

1.2%

0.787

1.4%

Notes: GRS: global radial strain; GCS: global circumferential strain; GLS: global Longitudinal strain; ICCs: intraclass correlation coefficients; CV: coefficient of variation.

Reviewer 3 Report

-Dear Authors it is a very interested and innovative study, very well designed with clear  and useful results.

-Feature tracking based on computed tomography imaging is an accurate method to evaluate and quantify myocardial strain. By evaluating the normal ranges of myocardial strain we can correlate them with cardiac function parameters.

-Detailed and clear criteria for excluding patients. Satisfactory number of healthy patients derived from an excessive list of patients in whom the examination was performed.

-All parameters of myocardial strain were evaluated ( GRS, GLCS, GLS)  and conclusions were drawn in a variety of characteristics sush as the gender, segments ,levels and regions.

-The presentation of the results, the tables and blueprints are very understandable.

-Comprehensible and explanatory writing, usage of English language in a very common and understandable way. 

-Use of adequate literature confirming the results.

-Clear differentation of the results by patient categories. 

-Very enjoyable to read.

Author Response

Points:

-Dear Authors it is a very interested and innovative study, very well designed with clear  and useful results.

-Feature tracking based on computed tomography imaging is an accurate method to evaluate and quantify myocardial strain. By evaluating the normal ranges of myocardial strain we can correlate them with cardiac function parameters.

-Detailed and clear criteria for excluding patients. Satisfactory number of healthy patients derived from an excessive list of patients in whom the examination was performed.

-All parameters of myocardial strain were evaluated ( GRS, GLCS, GLS)  and conclusions were drawn in a variety of characteristics sush as the gender, segments ,levels and regions.

-The presentation of the results, the tables and blueprints are very understandable.

-Comprehensible and explanatory writing, usage of English language in a very common and understandable way. 

-Use of adequate literature confirming the results.

-Clear differentation of the results by patient categories. 

-Very enjoyable to read.

Response: We are very grateful for your comments and time spent on the manuscript, we will continue to work hard in research.

Round 2

Reviewer 1 Report

Thanks for addressing the raised concerns